# 3′IsomiR Species Composition Affects Reliable Quantification of miRNA/isomiR Variants by Poly(A) RT-qPCR: Impact on Small RNA-Seq Profiling Validation

**DOI:** 10.3390/ijms242015436

**Published:** 2023-10-21

**Authors:** Adriana Ferre, Lucía Santiago, José Francisco Sánchez-Herrero, Olga López-Rodrigo, Josvany Sánchez-Curbelo, Lauro Sumoy, Lluís Bassas, Sara Larriba

**Affiliations:** 1Human Molecular Genetics Group—Bellvitge Biomedical Research Institute (IDIBELL), 08908 Hospitalet de Llobregat, Spain; aferre@idibell.cat (A.F.); luciasantiagolamelas1@gmail.com (L.S.); 2High Content Genomics and Bioinformatics (HCGB), Germans Trias i Pujol Research Institute (IGTP), 08916 Badalona, Spain; jsanchez@igtp.cat (J.F.S.-H.); lsumoy@igtp.cat (L.S.); 3Laboratory of Andrology and Sperm Bank, Andrology Service-Puigvert Foundation, 08025 Barcelona, Spain; olopez@fundacio-puigvert.es (O.L.-R.); jsanchez@fundacio-puigvert.es (J.S.-C.); lbassas@fundacio-puigvert.es (L.B.)

**Keywords:** miRNA isoforms, isomiRs, poly(A)-RT-qPCR, selective amplification, small RNAseq validation strategy

## Abstract

Small RNA-sequencing (small RNA-seq) has revealed the presence of small RNA-naturally occurring variants such as microRNA (miRNA) isoforms or isomiRs. Due to their small size and the sequence similarity among miRNA isoforms, their validation by RT-qPCR is challenging. We previously identified two miR-31-5p isomiRs—the canonical and a 3′isomiR variant (3′ G addition)—which were differentially expressed between individuals with azoospermia of different origin. Here, we sought to determine the discriminatory capacity between these two closely-related miRNA isoforms of three alternative poly(A) based-RT-qPCR strategies in both synthetic and real biological context. We found that these poly(A) RT-qPCR strategies exhibit a significant cross-reactivity between these miR-31-5p isomiRs which differ by a single nucleotide, compromising the reliable quantification of individual miRNA isoforms. Fortunately, in the biological context, given that the two miRNA variants show changes in the same direction, RT-qPCR results were consistent with the findings of small RNA-seq study. We suggest that miRNA selection for RT-qPCR validation should be performed with care, prioritizing those canonical miRNAs that, in small RNA-seq, show parallel/homogeneous expression behavior with their most prevalent isomiRs, to avoid confounding RT-qPCR-based results. This is suggested as the current best strategy for robust biomarker selection to develop clinically useful tests.

## 1. Introduction

Small non-coding RNAs (sncRNAs), with size <200 nucleotide (nt), are involved in the regulation of the expression of many genes and processes inside the cell (reviewed in [1]). The alteration of the sncRNA profile is associated with the development and progression of diseases, therefore, quantification of specific sncRNAs could reflect determinant information of specific pathological processes. Different RNA biotypes have been identified within sncRNAs which include microRNAs (miRNAs ~22 nt), PIWI-interacting RNAs (piRNAs 23–35 nt), endogenous interfering RNAs (endo-siRNAs 18–24 nt), transfer RNA (tRNA)-derived small RNAs (tsRNAs 18–40 nt), and ribosomal RNA (rRNA)-derived small RNAs (rsRNAs ~20 nt). In recent years the scientific community has worked hard to find the best method for their detection and quantification in biological tissues, cells and/or fluids; this issue has been very challenging due to their small size.

Small RNA sequencing (small RNA-seq) NGS-based technology can identify sncRNA fingerprints associated with physiological and/or pathological processes. To validate NGS data, the RT-qPCR approach is commonly used. As sncRNAs are typically shorter than 40 pb, the design of primers and PCR procedure is challenging. In order to overcome this limitation, several methods have been developed and adapted to commercially available kits, mostly for miRNA quantification and predominantly for miRNA canonical sequences (the referenced sequences at miRBase), but there has been a lack of attention on sncRNA-naturally occurring variants such as isomiRs. IsomiRs are miRNA isoforms, presenting substantial length and/or sequence heterogeneity, which are highly abundant (contributing to half of the miRNome in human cells) and can be expressed in a cell-specific manner [2,3,4]. These variants contribute as alternatively processed miRNAs to regulate target pathways in a coordinated manner [5]: whereas 5′variants can be affected in their seed sequence leading to different mRNA targeting (increasing the number of genes potentially regulated by a specific locus), and 3′isomiRs regulate common biological pathways. In this context, isomiRs are presumed to play a role in disease etiology [6]. Therefore, accurate quantification of isomiR abundance may be decisive.

The potential usefulness of miRNAs as non-invasive biomarkers in diagnosis and prognosis of human diseases has led to their extensive evaluation by the scientific community with promising results. Azoospermia, characterized by the absence of sperm in ejaculate, is a relatively common form of male infertility which may originate from a testicular spermatogenesis impairment—secretory azoospermia (SA) with no or few sperm in the testicle—or an obstruction in the genital tract—obstructive azoospermia (OA) with conserved spermatogenesis. The chance of sperm retrieval from testicular biopsy is high in cases with obstructive azoospermia, so that discriminating the origin of azoospermia is of great interest for assisted reproduction treatments. Previously we showed that miR-31-5p is abundant in semen exosomes and could serve as a marker for the origin of azoospermia [7,8]. Recently, using small RNA-seq we described miRNA/isomiR profiles in small extracellular vesicles (sEVs) from semen to be used as biomarkers for azoospermia [9]. Specifically, among those, we found that two isomiRs derived from the miR-31-5p locus were differentially expressed in semen sEVs in azoospermia of different origin [9]: one of them corresponded to the canonical sequence (AGGCAAGATGCTGGCATAGCT) and the other corresponded to a 3′isomiR variant—iso_3p:+1 (MiRTop nomenclature [3]; 3′ G addition). Interestingly, in the small RNA-seq study, the 3′isomiR variant showed higher expression than the canonical sequence, both in patients and in controls. Given the small size of these miRNA isoform sequences and their sequence similarity, the choice of validation assay has a great impact, because most RT-qPCR methods have not been designed considering isomiR variation. In general, it is assumed that qPCR assays based on polyadenylation reverse transcription—poly(A) RT—may give more reliable results compared to those based on the stem-loop RT [10].

The miRCURY LNA miRNA PCR system (Qiagen) is considered one reliable miRNA RT-qPCR commercially available technology to validate small RNA-seq results, and it is based on the poly(A) tailing of the miRNAs and a subsequent universal reverse transcription. PCR is performed with two specific locked nucleic acid (LNA)-spiked primers which have increased Tm and specificity (Busk PK, patent WO2010085966A2: Method for quantification of small RNA species). Although this method possesses high sensitivity for isomiRs that differ on internal base pairs, it is described often to exhibit lower reliability in quantification of individual isomiRs when the mismatch occurs at the 5′/3′ terminus [11], and thus, their discrimination is not guaranteed. To overcome this issue, other alternative poly(A) RT-qPCR based methods have been developed. Among those, miRPrimer2 technology [12] has been described to achieve the same specificity with DNA primers with optimized melting temperatures, and to detect and precisely quantify not only canonical miRNA but also isomiR sequences with a single base difference [13]. Alternatively, modifications of poly(A) RT-PCR systems (such as miR-X_Clontech) have been designed to increase selectivity to detect related isoforms [14].

Here we aimed at, first, determining the discriminatory capacity of the Qiagen miRCURY LNA miRNA PCR system, the miRPrimer RT-qPCR based method, and the modified Clontech miR-X poly(A) RT-PCR protocols to detect the two closely related isomiRs of the miR-31-5p locus, using synthetic RNAs and cDNAs, and secondly at measuring the isomiR quantification correlation between these RT-qPCR based methods and the high throughput RNA sequencing with biological samples. This aim is conceived as an essential step for the proper interpretation of RT-qPCR results when studying miRNA data in a biological context.

## 2. Results

In a previous study, we performed a high throughput small RNA-seq study in semen sEVs from individuals with azoospermia of different origin (OA versus SA) [9]. A signature of isomiR variants (including consensus and isoform miRNA sequences) was described to be differentially expressed between both groups of infertile men. Among these, several canonical miRNAs were tested by RT-qPCR with the miRCURY LNA miRNA PCR system (Qiagen) in a larger number of samples to validate their use as biomarkers for azoospermia [9]. However, various issues remain to be resolved. First, as this commercial kit was not designed to detect or quantify isomiR variants, our isomiR expression data still wait for validation. Additionally, the potential cross-reactivity of RT-qPCR assays among these very similar sequences (canonical and isomiR variants) is unknown, which could affect the reliability of the quantification results in terms of it reflecting exclusively the canonical miRNA expression or including that of other variants. 

In the present study, we focused on the quantification of two differentially expressed sequences between SA and OA individuals, from the same locus (Table 1: the hsa-miR-31-5p canonical sequence (21 bp) and the hsa-miR-31-5p_iso_3p:+1 3′ isomiR variant (3′ G addition; 22 bp). The small RNA sequencing data analysis showed 5-fold lower expression levels of the canonical sequence compared with the isomiR variant, both in azoospermic patients and fertile control individuals [9] (Appendix A). Thus, we applied specific assays, designed for each of the two isomiR species of this particular miRNA, and tested their discriminating ability by three alternative poly(A) based-RT-qPCR strategies: the commercial miRCURY, the miRPrimer2, and the modified miR-X RT-qPCR.

### 2.1. Assaying One Synthetic IsomiR at a Time

In order to determine the sensitivity of the three poly(A) based-RT-qPCR methods, we first tested serial dilutions of cDNA obtained from synthetic RNA. Standard curves for hsa-miR-31-5p and hsa-miR-31-5p iso_3p:+1 synthetic RNAs were assayed by RT-qPCR. A cDNA dilution series ranging from 10 to 10^8^ molecules was prepared for each qPCR reaction which showed an excellent linearity between the log of miR copy number per RT reaction and quantification cycle (Cq cycle) (R^2^: 0.99) with all three RT-qPCR strategies (Figure 1). No-template control reactions showed undetectable Cq values suggesting that the amplification values were obtained from interaction of DNA primers and cDNA obtained from synthetic RNA.

In the case of miRPrimer2 strategy, we also confirmed this result by using synthetic cDNA in the qPCR reaction. We built standard curves with serial dilutions of synthetic cDNA for the two isomiRs (Appendix A). Similarly, we were able to discriminate different dilutions of each isomiR, observing that the Cq value for each step in the serial dilution was consistently increased (R^2^: 0.99).

### 2.2. Assaying the Two Synthetic IsomiRs Simultaneously

In order to determine the specificity and discriminatory capacity of the qPCR reaction for the three strategies, we combined the cDNA (both, cDNA resulting from the synthetic RNA after RT reaction or the synthetic cDNA) of the two isomiRs into six pools with different proportion of both isomiRs (miR-31-5p/miR-31-5p iso_3p:+1): 0%/100%; 25%/75%; 50%/50%, 75%/25%, 100%/0% (Table 2. We chose the solution containing a concentration of 10^4^ molecules from the standard curve to obtain the different combinations. The expected Cq value was calculated from solutions containing the different percentages of one of each isomiR alone. This strategy allowed us to determine whether qPCR strategies could detect and properly quantify specific isomiR variants in the presence of similar sequences, being able to discriminate isomiRs present at different concentrations, as can be found similarly in fluids in physiological conditions.

A single peak was obtained in the melting curve of every isomiR mixture qPCR reaction, both with the consensus and the 3′ isomiR specific assays, with any of the RT-qPCR strategies (Appendix A). However, in Table 2 we show that the observed Cq values differed from those that we would have expected for amplification. Additionally, a considerable cross-reaction between 3′miRNA isoforms was shown, indicating that these poly(A) RT-qPCR methods are not able to exclusively detect a given isomiR. Cross-reactivity is revealed by the fact that the two synthetic isomiRs are detected with both assays designed for each of the specific miR-31-5p isoforms. Thus, if one were to attempt to examine isomiR levels using the isomiR specific primers, in order to confirm its absence of a sample, a background signal (presumably derived from the closely related miRNA isoform sequence) would be preventative in obtaining a reliable result. Conversely, to confirm the presence of a specific isomiR, one should be sure that the signal obtained from the RT-qPCR should detect only the specific sequence and not closely related sequences. When using the assays designed for amplification of the shorter (canonical) sequence, similar Cq values resulted in the different mixtures with all the three RT-qPCR strategies (Table 2A), suggesting that these amplify equally both the canonical and the 3′isomiR sequence. Although the assays designed for the longer (3′isomiR) sequence led to an increased selectivity, although insufficient, of amplification compared to assays targeting the shorter isoform (Table 2B), we showed that closely related isomiR variants contribute to the resulting qPCR signal, suggesting these three strategies are not able to precisely discriminate closely related isomiRs that are present with differences in the abundance between samples.

### 2.3. Assaying the Two IsomiRs in Biological Samples

Results from the three RT-qPCR strategies, showed more similar raw and normalized expression values between the canonical and the 3′isomiR variant for each of the samples (Figure 2 and Appendix A), compared with the small RNA-seq data, which revealed a 5-fold higher expression of the 3′isomiR variant than the consensus sequence, both in patients and in controls (Appendix A).

When comparing SA and OA semen sEV samples and referring to miRCURY RT-qPCR strategy results, statistically significant differences in the consensus miR-31-5p and also in the isomiR-31-5p variant expression levels were found (fold-change > 2; *p*-value < 0.001 Mann–Whitney U-test) (Figure 2A,B and Appendix A). The expression values of these sncRNAs resulted in good predictive accuracy, with the canonical miR-31-5p AUC and Sn producing slightly lower values (hsa-miR-31-5p: AUC 0.780, Sn 80.6, Sp 44.4) (hsa-miR-31-5p_iso_3p:+1: AUC 0.923, Sn 88.9, Sp 77.8). Similarly, when considering only the naturally occurring OA-N samples, miR-31-5p consensus and 3′isomiR variants resulted in being differentially expressed between OA-N and SA samples (hsa-miR-31-5p: AUC 0.811, Sn 97.1, Sp 14.3) (hsa-miR-31-5p_iso_3p:+1: AUC 0.899, Sn 94.1, Sp 57.1).

Similar results from SA vs OA group comparison were obtained when miRPrimer2 RT-qPCR strategy was used (Figure 2C,D and Appendix A): *p*-value < 0.005 (Mann–Whitney U-test) (hsa-miR-31-5p: AUC 0.753, Sn 97.1, Sp 44.4) (hsa-miR-31-5p_iso_3p:+1: AUC 0.752, Sn 79.4, Sp 44.4). Also, when considering only the naturally occurring OA-N samples, miR-31-5p consensus and isomiR variants resulted in being differentially expressed between OA-N and SA samples (hsa-miR-31-5p: AUC 0.903, Sn 97.1, Sp 57.1) (hsa-miR-31-5p_iso_3p:+1: AUC 0.794, Sn 97.1, Sp 14.3).

Conversely, despite obtaining a similar expression profile (Figure 2E,F and Appendix A), no significant result was obtained when using modified miR-X RT-qPCR strategy: *p*-value > 0.05 (Mann–Whitney U-test) and the predictive accuracy to select OA samples did not reach diagnostic efficiency (AUC < 0.7) (hsa-miR-31-5p: AUC 0.629, Sn 100, Sp 0) (hsa-miR-31-5p_iso_3p:+1: AUC 0.628, Sn 94.1, Sp 57.1).

## 3. Discussion

Recent results from our lab demonstrated quantitative changes in the miRNA/isomiR levels contained in seminal sEVs as clinically relevant, suggesting the use of extracellular vesicle miRNA/isomiRs as diagnostic molecular biomarkers in semen, which makes it possible to non-invasively diagnose the origin of azoospermia and the presence of spermatozoa in the testis in severe cases of male infertility [9]. Similar to other reports using miRNA deep sequencing, our previous results on semen EV sncRNA profiling showed multiple isomiR variants (3′isomiRs being more prevalent), of which, most of them, are more abundant than their canonical counterparts [9]. These miRNA isoforms with heterogeneous 3′ends are described to be post-transcriptionally processed by terminal nucleotidyltransferases (TENTs) which regulate miRNA sequences by the specifical addition of one or a few nucleotides to their 3′ ends [15,16,17]. In particular, TENT2 contributes to adenylation, guanylation, and uridylation on mature miRNAs whereas TUT4 uridylates most miRNAs [18]. The addition of nucleotides modulates the stability or degradation of miRNAs, contributing to mRNA expression pattern regulation.

Specifically, our previous small RNA sequencing data analysis showed altered expression levels of 185 isomiRs in OA compared with SA. Among them, two isoforms of miR-31-5p attracted our attention as miR-31-5p was previously described as a semen sEV biomarker for the origin of azoospermia by our group [7]. As higher expression levels of the hsa-miR-31-5p_iso_3p:+1 isomiR variant compared with the hsa-miR-31-5p canonical sequence, both in azoospermic patients and fertile control individuals, were observed in the small RNA-seq study [9], we also thought its role as a biomarker was relevant to be addressed.

Commercial RT-qPCR assays available nowadays are primarily designed to amplify the canonical miRNA variants as described in the miRbase without having the isomiR variation in mind. However, our current knowledge of miRNA heterogeneity has led us to test the capability of different RT-qPCR approaches to distinguish among similar isomiR sequences. The ability of LNA and miRPrimer2 primers to discriminate miRNAs with a single base difference has been described to be high when the position of the single nucleotide mismatch is located inside the sequence [11,12], however, the discriminatory capacity between 3′-end isoforms of the same miRNA family is controversial [14]: LNA polyadenylation RT-qPCR assays, designed to target longer miRNAs, have been described to fail when detecting shorter isoforms [13,14] unless these differ in length by three or more nucleotides. In our hands, not only miRCURY but also miRPrimer2 and modified miR-X RT-qPCR strategies exhibit a low reliable quantification of individual isomiRs when there is a single base length difference occurring at the 3′ terminus. Although these modified strategies increase the amplification selectivity of 3′ isomiRs, we showed that, as it occurs similarly when using LNA RT-qPCR [11,14], the miRPrimer2 and modified miR-X, primers show cross-reactivity and can amplify close sequences (primers targeted towards one isoform could also amplify the 3′ isoforms that are closely related in length, especially those differing by one nucleotide in length), and thus, these non-templated addition variants contribute to the qPCR resulting signal. Due to this cross-reactivity, these strategies are not suitable for discriminating among 3′isomiRs. This has a very strong impact on the value of RT-qPCR based validation of miRNA selected by small RNA-seq and corroborates previous studies which suggested that the presence of miRNA length isoforms can influence the interpretation of RT-qPCR results [19]. Although none of the poly(A) RT-qPCR strategies tested can guarantee the specific 3′ endpoints of the isomiR target and the ability of quantification strategies to discriminate closely related isomiRs is variable, among the three RT-qPCR strategies tested, LNA-based miRCURY and miRPrimer2 RT-qPCR strategies led to very similar results.

Our results show that these polyadenylation RT-qPCR techniques are not specific enough to accurately distinguish 3′isomiR sequences with a single base difference of a miRNA family, showing significant cross-reactivity, not only when using the assays towards amplification of the shorter sequence but also with the assays with amplification of a longer sequence. Fortunately, when assessing their use as biomarkers for azoospermia, the levels of these two hsa-miR-31-5p isomiRs (preferentially originated from testis and/or epididymis [7]) in semen EVs are similarly affected in azoospermia, showing a correlation between the levels of the canonical sequence and the 3′ isomiR variant and decreasing their levels in OA due to the obstruction in the ducts or vas deferens [9]. Therefore, the quantification results of these miR-31-5p isomiRs by poly(A) RT-qPCR techniques, although they are not accurate in terms of discriminating specificity, are consistent with the results of the small RNA-seq study with regards to detecting overall differences in the major isomiR species present in the seminal exosomal fraction between clinical groups. In conclusion, given that the two isomiR variants show changes in the same direction, the better performing qPCR assays compared in this work could potentially be used for miR-31-5p based biomarker testing of the origin of azoospermia.

A typical workflow to identify potential biomarkers for diagnosis purposes is to look first into a high-throughput profile of small RNAs by small RNA-seq (capable of identifying individual isomiRs) and then “validate” the differential expression pattern of specific candidates by other techniques such as the RT-qPCR approach. The expression behavior between the canonical miRNA and its isomiRs is usually heterogeneous, thus the use of this workflow has led to some discrepancies in miRNA quantification, even for the same miRNA and sample. This is probably due to the limitations that the RT-qPCR platform presents to discriminate different isomiRs that impact on the interpretation of results, and preferentially for those miRNAs of which the canonical variant is not the predominant one and/or the number of miRNA variants is high (the reads generated from the canonical sequence represent a small proportion of the miRNA signals) because due to cross-reactivity the signal may come mainly from the most predominant isoform. All of these issues have led to discard the use of some potentially useful miRNAs. The low correlation between miRNAs and their isoform expression is typically ignored which can contribute to obtaining low correlations between small RNA-seq detection and RT-qPCR validation. Therefore, when basing the validation strategy on commercially available assays, it will be very important to select for validation as biomarker those miRNAs with predominant isoform and/or the total of isoforms presenting the same expression behavior as the canonical miRNA sequence in a specific phenotype and/or disease.

Detection of isomiRs with expression diverging from the canonical isoform is technically challenging, and these variants can compromise canonical microRNA detection and quantification. Expression changes within the miRNA isoform population could be associated with disease state, and thus in the future it will be essential to develop and adopt new methods adjusted specifically to their detection to avoid confounding results. Thus, other techniques should be designed for discriminative quantification of isomiR variants [20]. Examining miRNA deep sequencing data is strongly recommended prior to selecting one miRNA for validation as biomarker. Currently, small RNA seq is the only platform which can accurately give information of the entire repertoire of miRNA/isomiRs, their relative abundance, and their expression fold-changes in the presence of disease.

Our results suggest the relevance of taking into account small RNA-seq profiling information, not only miR but also isomiR fingerprint associated with a pathology, in order to select for biomarker validation those miRNAs that show parallel/homogeneous expression behavior with their most prevalent isomiRs, to avoid confounding results. Other authors have suggested that a deep-sequencing-based approach should be developed to estimate the abundance of closely related small RNA variants [21]. Until standardized low-cost specific single molecule assays become available, the small RNA-seq strategy represents a good starting point for researchers interested in studying miR/isomiR potential as biomarkers of disease. Additionally, as miR/isomiRs can act cooperatively to control functionally important genes that are critical to physiological function, it may provide a more in-depth understanding of disease. While 3′-end isomiRs share common mRNA targeting with canonical miRNA, they may also have different cellular functions through distinct intracellular localization [22].

In conclusion, poly(A) RT-qPCR assays exhibit cross-reactivity between closely related isomiR sequences, that could lead to misinterpretation of qPCR-based validation of small RNA-seq results. We suggest that special care should be taken when validating small RNA-seq results by RT-qPCR approaches until more specific methods to detect miRNA isoforms are designed. Small-RNA based approaches, can pinpoint canonical miRNAs that show parallel/homogeneous expression behavior with their most prevalent isomiRs, to be prioritized first for RT-qPCR validation. This may be suggested as the current best strategy for selection of more robust biomarkers and the development of tests with clinical utility.

## 4. Materials and Methods

### 4.1. Subjects of Study

Patients and controls participating in the study were selected from men referred to the Andrology Service of the Fundació Puigvert. The Ethics Committee of both centers (F. Puigvert and IDIBELL) approved the study. An informed consent document was signed by every participant.

A comprehensive range of clinical tests was performed in selected patients, including hormonal analysis, histological evaluation of testicular biopsies, and, in the majority of cases, TESE (Testicular Sperm Extraction) outcome results (Table 3) thus allowing those patients to be categorized undoubtedly as azoospermic individuals with obstructive (OA) or secretory (SA) origin. Semen specimens were obtained from 34 infertile men diagnosed with SA (no sperm in semen sample due to spermatogenic failure) or cryptozoospermia (<0.15 × 10^6^ sperm/mL) and 18 individuals with OA and preserved spermatogenesis including both successfully vasectomized men (OA-V; *n* = 11) and individuals presenting pathological naturally occurring obstruction in the genital tract (OA-N; *n* = 7); this included two cases of post-epididymal obstruction (congenital ejaculatory duct obstruction and congenital absence of vas deferens -CBAVD-) and two cases of intratesticular obstruction). Additionally, 10 normozoospermic individuals, including sperm donors, were studied as controls.

The results of sperm retrieval using TESE provided information of the overall spermatogenic status which enabled the classification of azoospermia into obstructive (TESE value > 0.2 × 10^6^ sperm/mL) or secretory (<0.02 × 10^6^ sperm/mL), as well as allowing SA subgroups to be defined by determining the presence—SA(Sp+); *n* = 12—or the absence—SA(Sp−); *n* = 14—of sperm in a testicular biopsy (Table 3. The concentration of spermatozoa was obtained directly after processing 100 mg of biopsy in 1 mL of medium, which represents the TESE value.

All semen specimens were analyzed according to World Health Organization recommendations [23].

### 4.2. Sample Collection and Processing

Semen samples were obtained by masturbation after 3–5 days of sexual abstinence. Samples were allowed to liquefy for 30 min at 37 °C. Subsequently, they underwent two rounds of centrifugation (1600× *g* for 10 min, then 16,000× *g* for 10 min) at 4 °C in order to remove cells, cellular debris, and apoptotic bodies from the biofluid, and to obtain seminal plasma (supernatant) as detailed in previous publications [7,24,25]. Collected seminal plasma (SP) was immediately stored at −80 °C until needed.

### 4.3. Isolation of Small EVs from Seminal Plasma

SP aliquots (200 µL) were subjected to filtration through a 0.22 μm filter in order to remove macromolecules. The filtrate fluid plus 9 mL of PBS underwent ultra-centrifugation at 100,000× *g* using a SW40 rotor for 2 h at 4 °C as described elsewhere [7,24,25] to sediment the sEVs. The resulting pellet was resuspended in 100 µL PBS and then treated with RNAse A (Qiagen NV; Hilden, Germany) (100 μg/mL final reaction concentration; 15 min at 37 °C) to degrade the residual RNA outside the vesicles, before being subsequently stored at −80 °C. Nanoparticle tracking analysis was used to measure EVs and was performed by NanoSight NS300 (Malvern Instruments Ltd.; Malvern, UK) [7,8].

### 4.4. Small RNA-Containing Total RNA Isolation

Total RNA was obtained from sEV suspension samples using the miRNeasy Micro Kit (Qiagen) as previously described [8]. RNA concentration was calculated by using the QUBIT fluorometer and the Quant-iT RNA Assay kit (5–100 ng/µL) (Invitrogen; Carlsbad, CA, USA). RNA quality was determined by evaluating the OD 260/280 nm ratio with a Nanodrop UV-Vis spectrophotometer (Thermo Fisher Scientific; Waltham, MA, USA).

### 4.5. sncRNA Quantification by RT-qPCR

Three different methods were used:

#### 4.5.1. miRCURY LNA miRNA PCR System (Qiagen)

Reverse transcription (RT) of 10 ng of synthetic RNA oligonucleotides (RNAse-Free HPLC purification, Merck_Sigma Aldrich) was carried out—using the miRCURY^®^ LNA^®^ RT Kit (Qiagen)—which was subsequently used as a cDNA template to carry out the standard curve experiment of serial dilutions of each miRNA isoform: hsa-miR-31-5p (21 nt): AGGCAAGAUGCUGGCAUAGCU; hsa-miR-31-5p_iso_3p:+1 (3′ addition G; 22 nt): AGGCAAGAUGCUGGCAUAGCUG (Table 1. The standard curve was generated by ten-fold dilutions (10^1^–10^8^ molecules) of cDNA input.

An amount of 50 ng of semen sEV- RNA in a 10 µL reaction, using the miRCURY^®^ LNA^®^ RT Kit (Qiagen), was retrotranscribed to obtain first-stranded cDNA specific for miRNA. For qPCR analysis, 4 µL of diluted cDNA (12×) was assayed in 10 µL PCR reactions containing miRCURY LNA SYBR^®^ Green PCR Kit (Qiagen), as previously described [7,8]. Each miRNA was assayed in duplicate on a LightCycler^®^ 96 Instrument (Roche; Switzerland). miRNA qPCR assay for the hsa-miR-31-5p (ref YP00204236, Qiagen) was used whereas, for the hsa-miR-31-5p_iso_3p:+1 sequence, the mmu-miR-31-5p assay from Qiagen (ref YP00205159) was used, as this mouse canonical miRNA sequence is identical to that of the human 3′isomiR (Table 1).

To correct for potential overall differences between the samples, target miRNA expression was normalized using the mean expression value of hsa-miR-30e-3p and hsa-miR-30d-5p, previously described to be the most stable assays in semen samples with and without spermatogenic failure [7]. The relative quantification (RQ) miRNA expression values were calculated using the 2^dCq^ strategy.

#### 4.5.2. miRPrimer2 Strategy

Synthetic RNA oligonucleotides were used to obtain the miRNA standard curves as described for miRCURY strategy. Additionally, a standard curve for each specific miRNA was performed with syntHETIc cDNA oligonucleotide (RNAse-free HPLC purification, Thermo Fisher Scientific) miRNA-specific sequence as previously described [12] (syntHETIc cDNA sequences are detailed in Table 1. tHe cDNA standard curve was obtained through a series of ten-fold dilutions (10^1^–10^8^).

Reverse transcription of 40 ng of SEMen sEV total RNA in a final volume of 10 µL in the presence of ATP, RT primer (5′-CAGGTCCAGTTTTTTTTTTTTTTTVN, where V is A, C and G and N is A,C, G and T), poly(A) polymerase, dNTPs, and Superscript IV enzyme (200 U/µL) at 42 °C for 1 h as previously described [12]. cDNA samples were diluted (8×) and 10 µL PCR reactions were performed with 1 µL of diLUTEd cDNA, 5 µL of 2× SYBR Green mix (Roche), anD 250 nM of forward and reverse primers designed by miRprimer2 software version 2.0 (https://zenodo.org/record/1339289#.Ymj5i_exVGE, (accessed on 2 May 2022)) [12]. Sequences of primers are described in Table 1 Cycling conditions were set up as follows: 5 min at 95 °C; 40 cycles of 10 s at 95 °C, followed by 30 s at 60 °C, and finally melting curve analysis 60 °C to 99 °C. qPCR was also performed on a LightCycler^®^ 96 Instrument (Roche; Switzerland). The mean expression value of hsa-miR-30e-3p and hsa-miR-30d-5p was used for data normalization.

#### 4.5.3. Modified miR-X Poly(A) RT-qPCR Protocol

The Mir-X miRNA First-Strand Synthesis kit (Clontech) was applied on total RNA (25 ng) or synthetic RNA (to obtain the miRNA standard curves as described for the miRCURY and miRPrimer strategies) according to the manufacturer’s instructions, in a final volume of 10 µL in the presence of mRQ enzyme. The reaction took place at 37 °C for 1 h. cDNA samples were diluted (40×) and 10 µL of PCR reaction including specific miRNA forward primer (the entire sequence of the miRNA isoform which included 4A) added to the 3′-end as previously described [14]; Table 1, the mRQ 3′ primer (Clontech) as reverse primer and 4.6 µL of diluted cDNA. Cycling conditions were identical to the ones used for miRPrimer2 strategy on the LightCycler^®^ 96 Instrument (Roche; Switzerland).

### 4.6. Statistical Analysis

The non-parametric Mann–Whitney U-test was used to analyze differences in RQ miRNA expression levels between groups. Assessment of the ability of each small RNA tested to distinguish the samples with SA from those with OA was determined by performing a receiver operating characteristic (ROC) curve analysis of the RQ values. A binary logistic regression analysis (enter method) was used for predicting the association of each variable with the origin of azoospermia. Accuracy was measured as the area under the ROC curve (AUC). The threshold value was determined by Youden’s index, calculated as sensitivity plus specificity-1. SPSS statistical software version 15 (SPSS, CA, USA) was used, and a *p*-value < 0.05 was considered statistically significant.

## Figures and Tables

**Figure 1 ijms-24-15436-f001:**
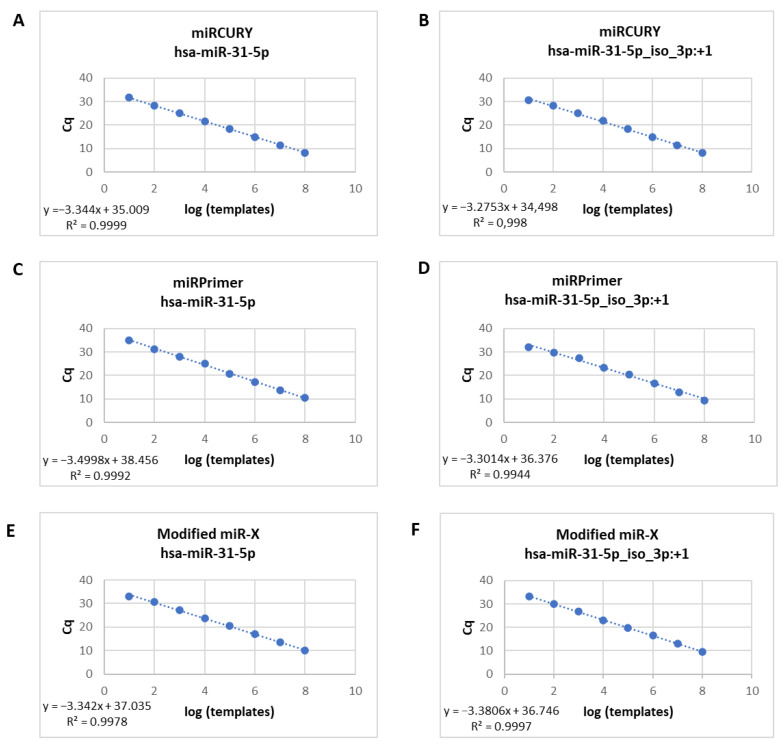
Standard curves for hsa-miR-31-5p and hsa-miR-31-5p iso_3p:+1 synthetic RNA assayed by RT-qPCR—the commercial miRCURY (**A**,**B**), miRPrimer2 (**C**,**D**) and modified miR-X (**E**,**F**) RT-qPCR strategies—in order to determine the sensitivity of these three RT-qPCR methods. Cq values extrapolated with serial dilutions of cDNA obtained from synthetic RNA exhibited outstanding linearity (R^2^: 0.99).

**Figure 2 ijms-24-15436-f002:**
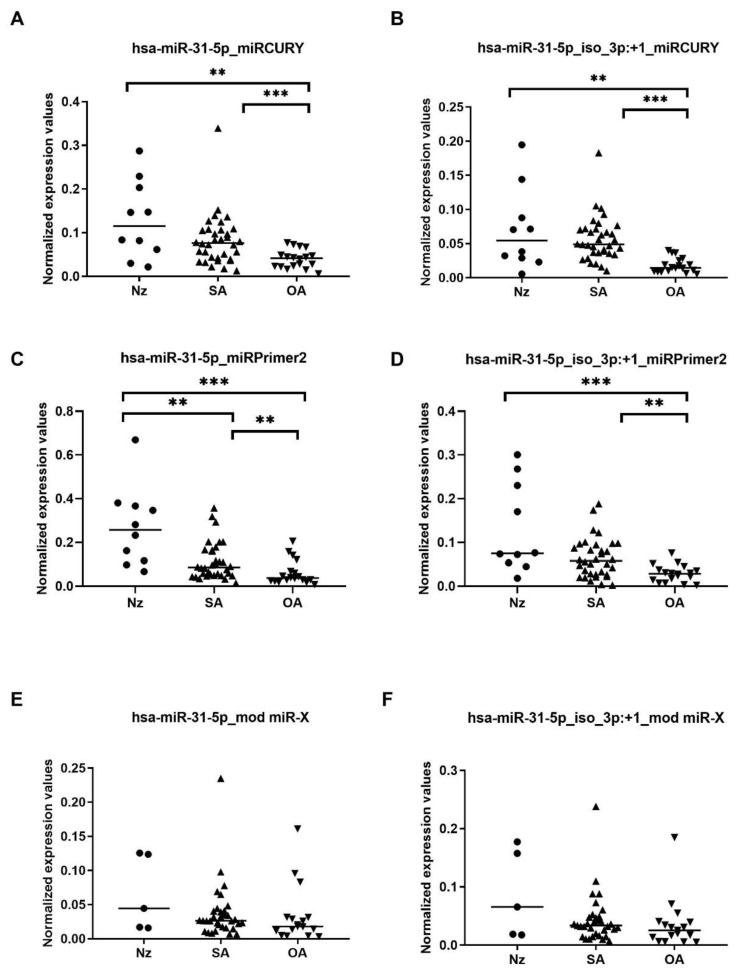
MicroRNA (miRNA) isoform levels in semen samples from azoospermia with different origin. Expression profiling of canonical hsa-miR-31-5p (**A**,**C**,**E**) and hsa-miR-31-5p_iso_3p:+1 (**B**,**D**,**F**) miRNA isoforms in seminal small extracellular vesicles (sEVs), obtained by reverse transcriptase-quantitative real-time polymerase chain reaction (RT-qPCR) quantification: miRCURY (**A**,**B**), miRPrimer2 (**C**,**D**), and modified miR-X (**E**,**F**) strategies. Data are shown as relative quantification (RQ) values, which were calculated using the 2^dCq^ strategy and relative to the expression values of (miR-30d-5p and miR-30e-3p) mean value. The horizontal bar displays the median expression value. Significant differences between groups are indicated: ** *p* < 0.01, *** *p* < 0.001 (Mann–Whitney U-test). Nz: normozoospermia (depicted as circle); SA: secretory azoospermia (depicted as triangle); OA: obstructive azoospermia (depicted as inverted triangle).

**Table 1 ijms-24-15436-t001:** miRNA isoform sequences, miRCURY assays, miRPrimer2 primers and synthetic cDNA templates; and miR-X primers.

		miRCURY	miRPrimer2			Modified_MiR-X
miRNA	RNA Sequence	miRCURY Assay	miRPrimer2 Forward Primer	miRPrimer2 Reverse Primer	miRPrimer2 Synthetic cDNA Template	Modified_MiR-X Forward Primer
hsa-miR-31-5p	AGGCAAGAUGCUGGCAUAGCU	hsa-miR-31-5p (ref YP00204236)	GCAGAGGCAAGATGCTG	GTCCAGTTTTTTTTTTTTTTTAGCTATG	CAGGTCCAGTTTTTTTTTTTTTTTAGCTATGCCAGCATCTTGCCT	AGGCAAGATGCTGGCATAGCTAAAA
hsa-miR-31-5p iso_3p:+1	AGGCAAGAUGCUGGCAUAGCUG	mmu-miR-31-5p (ref YP00205159)	GGCAAGATGCTGGCA	GTCCAGTTTTTTTTTTTTTTTCAGCTA	CAGGTCCAGTTTTTTTTTTTTTTTCAGCTATGCCAGCATCTTGCCT	AGGCAAGATGCTGGCATAGCTGAAAA
hsa-miR-30d-5p	UGUAAACAUCCCCGACUGGAAG	hsa-miR-30d-5p (ref YP00206047)	AGTGTAAACATCCCCGACT	GGTCCAGTTTTTTTTTTTTTTTCTTC		TGTAAACATCCCCGACTGGAAG
hsa-miR-30e-3p	CUUUCAGUCGGAUGUUUACAGC	hsa-miR-30e-3p (ref YP00204410)	GCAGCTTTCAGTCGGATGT	TCCAGTTTTTTTTTTTTTTTGCTGT		CTTTCAGTCGGATGTTTACAGC

Note: the RNA sequence that corresponds to miRPrimer forward primer is depicted in red whereas that corresponding to sequence complementary to reverse primer is depicted in blue.

**Table 2 ijms-24-15436-t002:** Assessment of isomiR discriminatory capacity of the three RT-qPCR strategies using assays designed for hsa-miR-31-5p (A) and hsa-miR-31-5p_iso_3p:+1 (B).

	miRCURY		miRPrimer2		Modified miR-X	
**(A) hsa-miR-31-5p assay**						
**miR|isomiR ratio (%)**	**Expected Cq values ^a^**	**Cq values**	**Expected Cq values ^a^**	**Cq values**	**Expected Cq values ^a^**	**Cq values**
**100**|0	22.50	22.09	25.23	25.32	24.32	24.06
**75**|25	22.79	22.11	25.60	25.44	24.81	23.97
**50**|50	23.55	22.11	26.18	25.55	25.41	24.00
**25**|75	24.81	22.08	27.33	25.53	26.35	23.58
**0**|100	36.35	22.04	-	25.69	-	23.86
**(B) hsa-miR-31-5p_iso_3p:+1 assay**						
**miR|isomiR ratio (%)**	**Expected Cq values ^a^**	**Cq values**	**Expected Cq values ^a^**	**Cq values**	**Expected Cq values ^a^**	**Cq values**
0|**100**	22.86	21.91	24.40	24.13	23.46	23.66
25|**75**	23.25	22.22	25.17	24.63	24.03	23.87
50|**50**	24.48	22.64	25.68	25.01	24.79	23.86
75|**25**	25.33	23.24	26.63	25.49	26.00	24.03
100|**0**	35.57	24.45	-	26.50	-	24.31

^a^ The expected Cq value was calculated from solutions containing the different percentages of only one of each isomiR.

**Table 3 ijms-24-15436-t003:** Clinical data of individuals included in the study.

Number of Patients (*n*)	Spermiogram	Subgroups	Age ^a^ (Years; Mean ± SD)	Sperm Count (×10^6^/mL)	Motility (%)	FSH (IU/L)	Testes Volume (mL; Range)	TESE Value (Pos/Neg; 10^6^ Sperm/mL)
10	Nz	Nz	29.50 ± 7.71	89.30 ± 57.96	49± 16.02	7.61 ± 5.08	18-22	ND
8	AZO	SA	33.50 ± 7.23	0	-	17.45 ± 11.62	4-20	ND
12	AZO	SA (Sp+)	40.25 ± 6.45	0	-	24.23 ± 12.71	1-18	Positive (<0.02)
14	AZO	SA (Sp−)	35.93 ± 4.79	0	-	23.63 ± 10.67	1-18	Negative
7	AZO	OA-N	39.14 ± 5.27	0	-	4.67 ± 2.47	15-20	Positive (>0.02)
11	AZO	OA-V	39.85 ± 8.35	0	-	ND	15-20	ND

Nz, normozoospermia; AZO, azoospermia and cryptozoospermia; OA-N, obstructive azoospermia due to pathological naturally occurring- obstruction in the genital tract; OA-V, obstructive azoospermia as a result of a vasectomy; SA, secretory azoospermia/cryptozoospermia; SA (Sp+), individuals with a positive TESE value; ND, not determined; TESE, testicular sperm extraction; pos, positive; neg, negative. ^a^ Age at the time of clinical assessment.

## Data Availability

Data can be provided on request.

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
