# Peer review of "3′IsomiR Species Composition Affects Reliable Quantification of miRNA/isomiR Variants by Poly(A) RT-qPCR: Impact on Small RNA-Seq Profiling Validation"

_ijms, 2023, doi:10.3390/ijms242015436_

Round 1
Reviewer 1 Report
The miRNA isoforms validation by RT-qPCR is challenging due to their small size and the sequence similarity. The authors used the previously sequenced miR-31-5 pisomiRs [the canonical and a 3’isomiR variant (3' G addition)] to perform RT-PCR validation tests. They sought to determine the discriminatory capacity between these two closely-related miRNA isoforms of three alternative poly(A) based-RT-qPCR strategies in both synthetic and real biological context. However, these poly(A) RT-qPCR strategies exhibit a significant cross-reactivity between these miR-31-5p isomiRs which differ by a single nucleotide, compromising the reliable quantification of individual miRNA isoforms. Therefore, the authors need to further optimize the experimental methods and procedures until satisfactory results are obtained.
The SSO in Table 1 has only two samples and how to calculate the standard deviation?
Reviewer 2 Report
In the manuscript "3'IsomiR species composition affects reliable quantification of miRNA/isomiR variants by poly(A) RT-qPCR: Impact on small RNA-seq profiling validation", Ferre et al. compared the expression of miR-31-5p and its isomiR variants in small extracellular vesicles from seminal plasma of patients with azoospermia of different origin with the aim to find novel diagnostic markers as well as to confirm the suitability of poly(A) RT-qPCR methods for NGS data validation. Importantly, they observed high cross-reactivity between individual isoforms of miR-31-5p which differ by only single nucleotide, thus miRNA selection for PCR validation should be performed with care as commercial assays are usually designed for canonical forms and the presence of different isoforms may lead to confounding results and discrepancy between NGS and validation data. In case of miR-31-5p and its most common 3'isomiR variant (3' G addition) in azoospermia patients, the authors observed the changes in expression in the same direction, thus the results of NGS were successfully confirmed during the validation phase although the canonical miRNA was not the most prevalent.
The topic of the paper is very interesting, original, and actual and could be interesting to the readers. The results are clearly presented and the quality of English is very good. I have only few minor comments or questions:
1) In the abstract - sncRNA - abbreviation should be introduced
2) Page 2 - sncRNAs are typically smaller than 40bp long - the combination of smaller and long is strange ... maybe they are shorter than 40bp?
3) Tables - all Cq values should have the same number of decimal places (two)
4) The number of patients in SSO group is very low to make some conclusions about the expression of miR-31-5p in these patients. Was it not possible to involve higher number of patients?
5) Page 9, last paragraph - ...isoforms are designed. Small-RNA based approaches can pinpoint (without two dots and comma)
6) Methods - What was the concentration of isolated RNA that it was possible to measure it on Nanodrop and to use quite high amounts for three different PCR methods used in the study?
7) Did the authors validate the quality and quantity of isolated small extracellular vesicles using DLS, EM, WB or any other methods?
8) Why did the authors use different amount of RNA when comparing three different PCR methods (50 ng in miRCURY, 40 ng in miRPrimer, 25 ng in miR-X)?
9) The authors demonstrated that the expression of the 3'isomiR variant is higher than the levels of the canonical variant. However, due to the high cross-reactivity during PCR and the fact that most assays are currently designed for canonical forms of miRNAs, they recommend canonical form of miR-31-5p to be selected for the validation phase of the study. However, have the authors also considered the possibility that the 3'isomiR variant may play a more significant role in the development of azoospermia and that its increased expression may have biological significance?
In conclusion, I recommend this paper to be published after minor revisions.
Round 2
Reviewer 1 Report
The miRNA isoforms validation by RT-qPCR is challenging due to their small size and the sequence similarity. The authors used the previously sequenced miR-31-5 pisomiRs [the canonical and a 3’isomiR variant (3' G addition)] to perform RT-PCR validation tests. They sought to determine the discriminatory capacity between these two closely-related miRNA isoforms of three alternative poly(A) based-RT-qPCR strategies in both synthetic and real biological context. The aim of this study was to to evaluate the discriminatory capacity of poly(A) RT-qPCR protocols to detect two closely related isomiRs, as an essential step for a proper interpretation of RT-qPCR results when studying miRNA data in a biological context. In this way, this paper is a good start and provides a good foundation and ideas for subsequent research.
Author Response
We thank the reviewer for his/her current positive evaluation of our manuscript. We have added a new phrase in the introduction to be better understood the aim of the study.
"Here, we aimed at determining the discriminatory capacity of the Qiagen miRCURY LNA miRNA PCR system, the miRPrimer RT-qPCR based method, and the modified Clontech miR-X poly(A) RT-PCR protocols to detect the two closely related isomiRs of the miR-31-5p locus, using synthetic RNAs and cDNAs, and at measuring the isomiR quantification correlation between these RT-qPCR based methods and the high throughput RNA sequencing with biological samples. This aim is conceived as an essential step for the proper interpretation of RT-qPCR results when studying miRNA data in a biological context."